# Time Course of Changes in the Neurovascular Unit after Hypoxic-Ischemic Injury in Neonatal Rats

**DOI:** 10.3390/ijms23084180

**Published:** 2022-04-10

**Authors:** Kazuki Hatayama, Sydney Riddick, Fares Awa, Xiaodi Chen, Daniela Virgintino, Barbara S. Stonestreet

**Affiliations:** 1Department of Pediatrics, Women & Infants Hospital of Rhode Island, Alpert Medical School, Brown University, Providence, RI 02905, USA; khatayama@wihri.org (K.H.); sydney_riddick@alumni.brown.edu (S.R.); fares_awa@alumni.brown.edu (F.A.); xchen@wihri.org (X.C.); 2Human Anatomy and Histology Unit, Department of Basic Medical Sciences, Neurosciences and Sensory Organs, University of Bari School of Medicine, 70124 Bari, Italy; daniela.virgintino@uniba.it

**Keywords:** blood-brain barrier, brain injury, hypoxia-ischemia, neonate, neurovascular unit

## Abstract

Exposure to hypoxic-ischemic (HI) insults in newborns can predispose them to severe neurological sequela. The mechanisms underlying HI-related brain injury have not been completely elucidated. The neurovascular unit (NVU) is a composite of structures that protect the brain from the influx of detrimental molecules. Changes in the NVU after HI are important because they could reveal endogenous neuroprotective pathways in the cerebral microvasculature. Furthermore, the time course of changes in the NVU after exposure to HI in the newborn remains to be determined. In this study, we examined the effects of severe HI on the time course of changes in the NVU in neonatal rats. Brains were collected from rats exposed to right carotid artery ligation and 2 h of hypoxia on postnatal day 7 with recovery for 6 or 48 h after exposure to sham treatment (Sham) or HI. The right HI and left hypoxic alone sides of the brains were examined by quantitative immunohistochemistry for vascular density (laminin), pericyte vascular coverage (PDGFRβ), astrocyte vascular coverage (GFAP), and claudin-5 expression in the microvasculature of the cerebral cortex, white matter, and hippocampus. HI-related brain injury in neonatal rats was associated with increases in vascular density in the cortex and hippocampus 48 h after HI as well as neurovascular remodeling, including loss of pericyte coverage in the cortex and increases in claudin-5 in the hippocampus 6 h after HI. Astrocyte coverage was not affected by HI injury. The time course of the responses in the different components of the NVU varied after exposure to HI. There were also differential regional responses in the elements of the NVU in response to HI and hypoxia alone.

## 1. Introduction

Hypoxic-ischemic (HI) brain injury can cause significant damage resulting in long-term neurologic sequelae such as cerebral palsy, sensory and/or motor impairment, developmental delay, and cognitive abnormalities [1,2]. The precise mechanism(s) fundamental to these abnormalities have not yet been completely elucidated, and only limited therapies are available to improve the outcomes after hypoxic-ischemic encephalopathy (HIE) in full-term infants [2].

The developing and adult brain are normally protected by a complex of barrier mechanism(s), which are termed the neurovascular unit (NVU) [3,4]. The NVU is a model in neuroscience, which represents modular structural and functional multicellular associations between the brain and blood vessels [3,4]. The NVU is composed of neurons, perivascular astrocytes, microglia, pericytes, endothelial cells (EC), and the basement membrane (BM). These components share intimate and complex associations and interactions with each other [3,4]. The NVU is responsible for the maintenance of the highly selective blood-brain barrier (BBB) and cerebral homeostasis, as well as the control of cerebral blood flow [4]. These barrier mechanisms are formed early in development and are important for establishing a favorable environment and facilitating nutritional support for all the different cell types during the maturation of the brain [5]. Therefore, the loss of integrity of these barrier mechanisms could potentially result in changes in developmental processes.

Previous work has shown that ischemia in the fetus and HI in neonatal rodents can increase BBB permeability and disrupt the BBB in part by altering some components of the NVU [1,6]. Studies have also shown disturbances in the NVU during ischemic insults in both adult and neonatal subjects [1,4,7,8]. In addition, recovery of some components of the NVU has been reported after stroke in the adult brain [8]. It is critical to understand the breakdown and repair processes of the NVU after HI-related insults because they could reveal potential endogenous neuroprotective pathways that could protect the cerebral microvasculature after exposure to HI. Although previous studies have reported alterations in the NVU after ischemia in the fetus, HI in neonatal rodents, and stroke in adult subjects [1,6,8,9], the time course of changes in the elements of the NVU after exposure of neonates to HI remains to be determined.

The overall goal of the current study was to investigate the time course of changes in selected structural components of the NVU in neonatal rats after exposure to severe HI. We examined the density of blood vessels in the brain, pericyte and astrocyte vascular coverage of microvessels, and claudin-5 expression in microvessels after recovery for 6 and 48 h from HI induced by right carotid artery ligation and exposure to 8 percent oxygen for 2 h to understand the time course of changes in selected elements of the NVU after exposure to severe HI.

## 2. Results

### 2.1. Microvessel Density

Laminin is a key molecular constituent of the vascular basal lamina. It has been extensively used as an excellent marker for the microvasculature network in rodents [10,11]. Therefore, laminin staining was used in the current study to visualize the microvascular network in the cerebral cortex, white matter, and hippocampus of the neonatal rat brains (Figure 1). Figure 1A contains the immunohistochemical images of microvessels stained with anti-laminin antibodies in the cerebral cortex, white matter, and hippocampus on the left hypoxic and right HI sides of the neonatal rat brains in the Sham, HI-6 h, and HI-48 h groups. The overall images are shown at a magnification of 10×. The boxed dotted insets show representative microvessels. The solid insets show an enlarged image of the microvessels. Figure 1B shows the quantified microvessel density expressed as laminin expression as a percent of the area stained with the anti-laminin-antibody in the sections of the cerebral cortex, white matter, and hippocampus on the left hypoxic and right HI sides of the brain in the neonatal rats. Exposure to HI was associated with increases (*p* < 0.05) in microvascular density 48 h after HI compared with the Sham group. The laminin expression on the right HI side of the brain was also higher than on the left hypoxic side in the cerebral cortex 48 h after HI (Figure 1A,B). However, differences in vascular density were not observed in white matter (Figure 1A,B). In addition, differences in vascular density were also observed in the hippocampus between the left hypoxic and right HI sides of the brain at 48 h after HI (Figure 1A,B).

### 2.2. Pericyte Microvascular Coverage

PDGFRβ (platelet-derived growth factor receptor beta) immunostaining has been widely used as a molecular marker for pericytes in rodents [12]. PDGFRβ staining was used in the current study to visualize the pericytes in the cerebral cortex (Figure 2A), white matter (Appendix A), and hippocampus of the neonatal rats (Appendix A) on the left hypoxic and right HI sides of the brain in the Sham, HI-6 h, and HI-48 h groups. The immunohistochemically stained sections (Figure 2A) show the PDGFRβ, laminin and merged images for the hypoxic left and HI right sides in the cerebral cortices of the Sham, HI-6 h, and HI-48 h groups. The overall images are shown at a magnification of 40×. The boxed dotted insets show representative microvessels covered by pericytes. The solid white insets show an enlarged image of the pericytes. The quantified vascular coverage of pericytes is shown as the percent of PDGFRβ vascular coverage in the cerebral cortex, white matter, and hippocampus as dot plot figures in the neonatal rats (Figure 2B). Pericyte coverage was significantly lower 6 h after exposure to HI compared with the Sham group (Figure 2B). Differences in pericyte coverage were not observed in white matter nor in the hippocampus on the hypoxic or HI sides of the brain (Figure 2B).

### 2.3. Astrocyte Coverage

GFAP (glial fibrillary acidic protein) staining was used to visualize astrocytic vascular coverage in the cerebral cortex (Appendix A), white matter (Appendix A), and hippocampus (Appendix A) of the left hypoxic and right HI sides of the brain in the Sham, HI-6 h, and HI-48 h groups. The quantified vascular coverage of the astrocytes in the cerebral cortex, white matter, and hippocampus is shown as the dot plots as the percent of vascular coverage plotted for the hypoxic left and HI right sides of the brain in the neonatal rats (Figure 3). Significant differences in quantified astrocytic vascular coverage were not observed in the cerebral cortex, white matter, or hippocampus on either side of the brain (Figure 3).

### 2.4. Claudin-5 Expression

Claudin-5 staining was used to visualize claudin-5 expression in the microvasculature as a representative of the tight junction proteins in the cerebral cortex (Appendix A), white matter (Appendix A), and hippocampus (Figure 4A) on the left hypoxic and right HI sides of the brain in the Sham, HI-6 h, and HI-48 h groups. Claudin-5, laminin, and the merged images are shown for the hippocampus on the left hypoxic and right HI sides of the brain in the Sham, HI-6 h, and HI-48 h groups (Figure 4A). The overall images are shown at 40× magnification. The dotted boxed insets show representative microvessels with the expression of claudin-5. The solid white boxed insets show enlarged images of claudin-5 within the microvessels. Quantified claudin-5 expression is shown as the percent of claudin-5 expression in the microvessels of the cerebral cortex, white matter, and hippocampus as the dot plots in the groups of neonatal rats (Figure 4B). The quantified claudin-5 expression in the cerebral cortex and white matter did not differ on the left hypoxic or right HI sides of the brain between the groups. However, the claudin-5 vascular expression in the hippocampus was significantly higher in the HI-6 h and HI-48 h groups compared with the Sham group on the left hypoxic side of the brain and was also higher in the HI-6 h compared with the Sham group on the right HI side of the brain (Figure 4B).

## 3. Discussion

HI-related brain injury can result in severe morbidity and mortality in the newborn [1,2,7,9]. Loss of NVU integrity is an early and prominent pathological feature of neuroinflammatory disorders and HI-related brain injury that can disrupt the homeostasis of the brain, induce cerebral edema, and facilitate entry of inflammatory elements into the CNS, which predispose to perinatal brain injury [9,13]. The purpose of the current study was to determine the potential changes in the NVU after HI-related brain injury in the neonatal rat brain to elucidate some additional mechanisms that could predispose an individual to brain injury during the perinatal period [9]. In this study, we quantitatively examined the immunohistochemical expression of the basal lamina, pericytes, astrocytes, and claudin-5 in the microvasculature of the cerebral cortex, white matter, and hippocampus from the hypoxic and HI sides of brains in Sham control animals that were not exposed to hypoxia or HI, and in animals exposed to hypoxia and HI with recovery from these insults for 6 and 48 h after HI-related brain injury.

The major findings of the current study were as follows. First, microvessel vascular density was increased in the right HI cerebral cortex 48 h after exposure to HI compared with the Sham treated group and with the left cerebral cortex exposed to hypoxia alone. Second, the microvessel density also increased in the right hippocampus 48 h after exposure to HI compared with the left hippocampus exposed to hypoxia alone. Third, pericyte coverage was decreased in the right cerebral cortex 6 h after exposure to HI compared with the Sham treatment. However, there were no differences in pericyte coverage in the right cerebral cortex 48 h after HI compared with the Sham group or on the left side that was exposed to hypoxia alone. Fourth, there were no differences in pericyte coverage in white matter or hippocampus after HI or hypoxia alone. Fifth, there were no differences in astrocyte coverage in the cerebral cortex, white matter, or hippocampus after HI. Sixth, claudin-5 expression in the microvasculature increased in the left hypoxic and right HI hippocampus 6 h after the insults compared with the Sham group and also in the left hypoxic hippocampus 48 h after the insults. In addition, there were no differences in the vascular claudin-5 expression in the cerebral cortex or white matter after the insults.

There is considerable evidence to suggest that pathophysiological events occurring during the perinatal period could alter the structure and function of the NVU [4]. These pathophysiological disorders include prematurity [14], uteroplacental inflammation [15,16], and HI-related brain injury [1]. The disruption of the NVU resulting from uteroplacental inflammation has been documented by the presence of plasma albumin extravasation outside of the cerebellar vasculature [15] and reduced microvascular density, lower pericyte vascular coverage, and decreased astrocyte vascular coverage [16]. HI-related insults to the developing brain have been reported to result in the transient opening of the BBB, which appeared to recover over time [1]. These changes were associated with reductions in tight-junction proteins as well as changes in the distribution of the proteins after HI. Changes in regional cerebral blood flow also exhibited correlations with the brain regions that were injured by HI and regions that exhibited barrier opening [1]. In the current study, we examined the time course of changes in the NVU after recovery from HI-related brain injury to elucidate the disruption and repair processes after exposure to HI and to determine potential endogenous changes in the cerebral microvasculature.

The microvasculature quantification was determined with anti-Laminin antibodies. Laminin is an important molecular component of the vascular basal lamina, which has been extensively used as an important marker of the microvasculature in rodents [10,11] and other species, including humans [10,17,18,19,20,21].

In the current study, microvessel density was increased 48 h after HI in the cerebral cortex and hippocampus. These findings are consistent with our previous work, in which we have demonstrated that the onset of neovascularization occurred between 48 and 72 h after brain ischemia in the cerebral cortex of the ovine fetus [22]. Neurogenesis and angiogenesis can occur in response to ischemic brain injury and could potentially attenuate brain injury [23]. In vitro evidence suggests that endothelial cells secrete active substances that promote neuronal survival and, thereby, facilitate neuroprotective effects after ischemic insults [24]. Consequently, cerebral neovascularization that appears to occur within 48 h after HI-related brain injury in the neonatal rat could potentially serve to ameliorate ischemic damage to the brain. Although hypoxia-induced angiogenesis is potentially neuroprotective because it could increase oxygen and nutrient delivery to the hypoxic-ischemic tissue, there also can be associated adverse effects such as alterations in the BBB and NVU function that are not necessarily beneficial [25].

Pericytes are critical cellular components of the NVU that are central to the development and maintenance of the blood-brain barrier because of their contributions to the regulation of the endothelial cellular junctions [12,26,27,28]. Pericytes contribute to angiogenesis and perform cellular immune functions in addition to their contribution to the structural integrity and stability of the NVU [29]. Coverage of the microvasculature by pericytes in the premature human cerebral cortex and white matter is approximately 90% when determined with the neural/glial antigen 2 (NG2) and platelet-derived growth factor receptor beta (PDGFRβ) immunohistochemical markers [30]. In the current study, we used anti-PDGFRβ antibodies to identify pericytes and showed that the pericyte coverage was approximately 90–100% in the cerebral cortex and hippocampus and approximately 80–100% in the white matter in the brain of Sham treated P7 neonatal rats, which is roughly similar to the findings in premature infants [30]. The coverage was reduced from 100% to 80% 6 h after exposure to HI-related brain injury in the HI right cortex compared with Sham-treated neonatal rats. These findings can be interpreted to suggest that abnormalities in the pericyte coverage could occur within a relatively short time frame after exposure to HI-related insults to the brain. There is a paucity of information regarding the role of pericytes after HI in the neonatal brain. Information regarding the response of pericytes to ischemic injury has been mostly reported in the adult brain. Pericytes have been demonstrated to migrate away from the cerebral microvessels after stroke in adult subjects [31,32]. These findings are consistent with the reductions in microvascular pericyte coverage 6 h after HI brain injury in the cerebral cortex of the neonatal rats. However, differences were not detected in microvascular pericyte coverage in the cerebral cortex 48 h after HI compared with the Sham treated group, suggesting that there could be a restitution of microvascular pericyte coverage within 48 h after HI insults to the brain in neonatal rats. Ek et al. have reported transient increases in blood-brain barrier permeability after HI in neonatal mice [1]. Therefore, taken together, considering the contribution of pericytes to the integrity of blood-brain barrier function, restitution of microvascular coverage by pericytes could potentially contribute to the recovery of blood-brain barrier function after HI injury in neonatal rodents [33,34]. Additionally, there were no differences in pericyte coverage in the white matter or hippocampus after HI or hypoxia alone, suggesting that there are regional differences in the responses of pericytes to these insults.

Astrocyte end-feet ensheathe the blood vessels in the brain and provide structural support to the cerebral vasculature [18]. Astrocytes are important for the development and maintenance of the blood-brain barrier, brain homeostasis, structural support, control of cerebral blood flow, and secretion of neuroprotective factors [35]. Astrocytes are activated within minutes after injury by pro-inflammatory mediators, cytokines, and reactive oxygen species that are secreted by injured neurons and glial cells [36]. One unique role of astrocytes in HIE-induced inflammatory responses is that, in addition to the release of cytokines and chemokines, reactive astrocytes have the ability to up-regulate the expression of inflammatory mediators in neuroblasts and angioblasts in neonatal brains, which are chemotactic for bone marrow-derived immune cells [37]. Astrocyte coverage in the developing human brain is approximately 60% when measured with glial fibrillary acidic protein (GFAP) [18].

Many published studies have previously used GFAP as a marker for astrocytes, including our own studies [16,17,18,22,38]. Nevertheless, previous studies have also used alternative markers, including N-myc-downregulated gene2 (NDRG2) and S100b, to detect the distribution and number of astrocytes [18,39,40,41,42]. Therefore, it would be of great interest to verify and compare the developmental and regional distribution and the effects of HI-related brain injury on these various astrocytic markers in future experimental paradigms. On the other hand, there may not be one single optimal marker for astrocytes under all experimental conditions, developmental ages, and in all brain regions [41]. Nonetheless, we have previously used this identical method to quantify astrocytic coverage in fetal sheep cerebral vasculature with and without exposure to LPS and were able to detect significant differences in coverage between the groups with decreased vascular coverage in white matter and similar non-significant patterns in the cerebral cortex of the fetal sheep [16]. Consequently, we have used GFAP as a marker for astrocytes in our study.

In the current study, we showed that the astrocyte coverage was approximately 60–80% in the cerebral cortex, 70 to 80% in the white matter, and 70% to 90% in the hippocampus in the Sham control rats. There were no significant differences in the astrocyte coverage after exposure to HI or hypoxia or at the different time points in any of the brain regions after HI. Experimental studies regarding astrocytic responses to HIE or systemic LPS exposure in fetuses from various species suggested that astrocytes are relatively resistant to injury during the neonatal period and that the astrocytes adjacent to regions of necrosis are poised to proliferate [36]. The current results could reflect the durability of the astrocyte population after HI-related brain injury during the neonatal period.

Tight junctions (TJs) of the BBB are present between the endothelial cells of brain capillaries [43]. There are two primary classes of proteins at the TJs. Transmembrane proteins, including occludin, claudins, and junctional adhesion molecules (JAMs), and peripheral proteins, including the zonula occludens family, AF6/afadin, multi-PDZ domain protein (MUPP1), membrane-associated guanylate kinase inverted (MAGI)-1, -2, and -3, PAR-3 and -6, and heterotrimeric G-proteins [43]. The degree of barrier tightness is determined by interactions between TJ family members and the endothelial cells [44]. Claudin-5 is the most enriched tight junction protein at the BBB, and its dysfunction has been implicated in neurodegenerative disorders and neuroinflammatory disorders [44]. We focused on the expression of claudin-5 in the current study because claudin-5 is considered a key factor involved in the endothelial permeability of the BBB [43,45]. The NF-κB/MMP-9 signaling pathways contribute to the degradation of claudin-5, prompting hemorrhagic transformation in ischemic stroke [45]. The expression of claudin-5 was increased after HI injury and hypoxic injury in the hippocampus in the present study. This is consistent with a previous report that showed the increases in claudin-5 protein and gene expression at 6 h after neonatal HI in the hippocampus and cortex [1]. The expression of claudin-5 can be affected by pericytes and astrocytes. Glial cell line-derived neurotrophic factor, a member of the TGF-β superfamily secreted from pericytes, increases the expression of claudin-5 in an in vitro human BBB model and enhances barrier properties [46]. In experimental autoimmune encephalomyelitis (EAE), loss of claudin-5 is a primary event in affecting the BBB disruption due to the overexpression of VEGF-A from glial GFAP positive astrocytes [47]. However, additional study is required to establish the relationship between glial cells and tight junction proteins after the HI-related brain injury during the neonatal period.

Recently, there has been increasing evidence that alterations in the components of the NVU could be a reasonable source of biomarkers as indicators for brain injury after stroke in adult subjects [48]. Furthermore, recent and past studies have reported that brain-derived neurotrophic factor (BDNF) has a critical role in ischemic brain injury and that it could also be a potential biomarker for ischemic and traumatic brain injuries [48,49]. In this context, although our current study focused on changes in the structural components of the NVU, it would be of interest to examine changes in BDNF and other growth factors with reference to changes in the NVU after HI-injury-related brain injury in the neonatal in future studies.

In summary, the results of the current study appear to suggest that the NVU exhibits durability and regenerative capacity in response to HI injury in the neonate. We demonstrated that the pericytes might recover after HI injury and that astrocyte coverage was not significantly reduced after the injury, and claudin-5 expression was increased after the injury.

There are several limitations to our study. Although we have identified the time course of changes in NVU components for the first time after HI injury in neonatal rats by immunohistochemical quantification, we were not able to complement these results by Western immunoblot analysis or enzyme-linked immunosorbent assay (ELIZA) on the brain samples from these studies because freshly frozen brain tissue was not available from the original studies. Moreover, we cannot comment upon sex-related differences due to the limited number of animals in the current study. In addition, we were not able to avoid the neuronal and nerve fiber staining, which could have been related to aldehyde fixative-induced auto-fluorescence because our samples required a long interval of fixation for optimal preservation after exposure to the 2 h of hypoxia-ischemia and the necessity to preserve the cellular and subcellular structures of the NVU. This could have affected the fluorescence quantitation.

In conclusion, the current study revealed the time course of changes in NVU after HI-related brain injury in neonatal rat brains for the first time. HI injury was associated with increases in vascular density and neurovascular remodeling, including the loss of pericyte coverage in the cortex and increases in claudin-5 in the hippocampus, in neonatal rats. The time courses in the responses of the different components of the NVU varied after exposure to HI. There were also differential regional responses in the elements of the NVU in response to hypoxia and HI. We speculate that these alterations in the NVU could predispose to neuronal injury in the neonatal brain.

## 4. Materials and Methods

### 4.1. Animal Preparation

The experimental procedures in this study were conducted after obtaining approval from the Institutional Animal Care and Use Committees of the Alpert Medical School of Brown University and Women and Infants Hospital of Rhode Island and in accordance with the National Institutes of Health Guidelines for the use of experimental animals. The brain tissue samples utilized in this study were residual samples obtained from neonatal rats enrolled in our previous report [50].

The neonatal rats were born in time-mated dams (Charles River Laboratories, Shrewsbury, MA, USA) in the Animal Care Facility of Brown University. Pregnant Wistar dams were transported on embryonic day 15 (E15) or E16 and housed in a 12 h light/dark cycled facility with ad libitum access to food and water. The date upon which the rat pups were delivered was designated as postnatal day zero (P0). The pups from different litters born on the same day were culled and balanced so that each dam had no more than 10 pups. The pups were randomly assigned to the sham-operated control and HI-exposed groups on day 7 (P7). HI was induced by right carotid artery ligation along with exposure to 8% oxygen for 2 h according to the previously described methods [50,51].

Briefly, the pups were anesthetized with 3% to 4% isoflurane and maintained with 1% to 2% isoflurane during the procedure. The temperature of the pups was maintained at 36 °C during surgery with an isothermal heating pad. A midline skin incision was made on the neck overlying the trachea using sterile surgical scissors. The right common carotid artery (RCCA) was separated from the trachea and surrounding nerves and double ligated with 5-0 silk sutures. The incision was closed and sterilized with betadine and alcohol. Sham-treated subjects were exposed to the same procedure, except that the RCCA was not ligated. The pups were identified with a neonatal tattooing system (Neo-9, Animal Identification & Marking Systems, Inc., Hornell, NY, USA). They were then returned to their dams for 1.5 to 3 h for feeding and recovery from the surgery and then were subsequently placed in a hypoxia chamber with 8% humidified oxygen and balanced nitrogen for 2 h with a constant temperature of 36 °C. The sham control-treated subjects were similarly exposed to room air for 2 h. The rats in each group were obtained from different litters to lower the potential for inter-litter variability. The animals in the Sham and HI groups were sacrificed at the same time points after exposure to sham HI or HI. The brains were obtained for the purpose of analysis from neonatal rats in the group of sham-operated control (Sham; *n* = 6 (male; *n* = 4, female; *n* = 2)), and in the group of neonatal rats at 6 h (HI-6 h, *n* = 6 (male; *n* = 3, female; *n* = 3)) and 48 h (HI-48 h, *n* = 9 (male; *n* = 4, female; *n* = 5)) after exposure to HI [50].

The study design is schematically shown in Figure 5. After the ligation of the right carotid artery, the neonatal rats were exposed to 8 percent oxygen for 120 min and allowed to recover for 6 and 48 h. The schematic of the coronal section of the neonatal rat brain shows the regions of the brain that were examined in this study.

Newborn pups from pregnant Wistar rats were used in this study. On postnatal day 7, HI was introduced by right carotid artery ligation along with exposure to 8% oxygen for 2 h. Brains were collected at 6 h and 48 h after HI injury. Images were randomly taken from the left hypoxic and right HI sides of the brain from different regions, including the cortex, white matter, and hippocampus.

### 4.2. Brain Collection and Immunohistochemical Staining

The brains were collected from rat pups after exposure to right carotid artery ligation and hypoxia (HI) with reoxygenation and recovery for 6 h (HI-6 h: *n* = 6) or 48 h (HI-48 h: *n* = 5) or sham control treatment (Sham: *n* = 6). The pups were sedated with intraperitoneal injections containing ketamine (74 mg/kg) and xylazine (4 mg/kg). A hind leg pinch was performed after the injections to ensure that there was adequate sedation. The rat pups were perfused with cold saline and 4% paraformaldehyde (PFA) at a flow rate of 3 mL/min through the left ventricle as previously described [50]. The brains were removed and fixed in PFA for 24 h. After fixation, the brains were processed and embedded in paraffin. Paraffin-embedded brain tissue was sectioned coronally at a thickness of 6-micron. A coronal section of brain tissue containing the dorsal hippocampus was used to standardize the analyses as previously described based upon the location of the bregma −3.12 ± 0.6 mm [52].

The slides were incubated in an oven at 60 °C for 1 h. Then, they were deparaffinated and rehydrated. The tissue sections were then heated and pressurized for 20 min in a citrate-based antigen unmasking solution (Vector Laboratories, Burlingame, CA, USA) for antigen retrieval. Then, they were placed in a humidified chamber and blocked with Superblock T20 Blocking Buffer (Thermo Fisher Scientific, Waltham, MA, USA) for 2 h after three washes with PBS at 5 min intervals. The slides were then incubated overnight at 4 °C with appropriate primary antibodies. The primary antibodies were as follows: rabbit anti-laminin polyclonal antibody (#L9393, Sigma-Aldrich, St. Louis, MO, USA) applied at a dilution of 1:200, mouse anti-PDGFRβ monoclonal antibody (#SC374573, Santa Cruz Biotechnology, Santa Cruz, CA, USA) administered at a dilution of 1:10, chicken anti-GFAP monoclonal antibody (#AB4676, Abcam, Cambridge, MA, USA) applied at a dilution of 1:500, and mouse anti-claudin-5 monoclonal antibody (#35-2500, Thermo Fisher Scientific, Waltham, MA, USA) applied at a dilution of 1:20. The next day, after three washes with PBS at 5 min intervals, the slides were incubated for 1 h with appropriate secondary antibodies. The secondary antibodies were as follows: biotinylated goat anti-rabbit antibody (#BA-1000-1.5, Vector Laboratories, Burlingame, CA, USA) applied at a dilution of 1:400, Alexa Fluor 594 goat anti-mouse antibody (#A32742, Thermo Fisher Scientific, Waltham, MA, USA) applied at a dilution of 1:1000, and Alexa Fluor 594 goat anti-chicken antibody (#A32932, Thermo Fisher Scientific, Waltham, MA, USA) applied at a dilution of 1:1000. Then, after three washes with PBS at 5 min intervals, the slides were incubated for 1 h with Streptavidin, Alexa Fluor 488 conjugate (#S11223, Thermo Fisher Scientific, Waltham, MA, USA), which was applied at a dilution of 1:300. Finally, the immunoreactions were visualized with DAPI (VECTASHIELD Antifade Mounting Medium with DAPI H-1200, Vector Laboratories, Burlingame, CA, USA).

### 4.3. Image Acquisition and Quantification

An ORCA-Flash4.0 LT+ Digital CMOS camera (Hamamatsu, Hamamatsu, Japan) connected to the computer software Stereo Investigator 10.0 (MicroBrightField, Inc., Williston, VT, USA) was used for image acquisition and subsequent quantification. Six fields from the cerebral cortex on each side of the brain were randomly selected and obtained at 10× magnification to determine the laminin density. Ten fields were collected at 40× magnification to examine vascular pericyte and astrocyte coverage and claudin-5 expression. In addition, two fields from the hippocampus from each side of the brain were randomly selected and obtained at 10× magnification to examine laminin density and seven at 40× magnification to examine vessel coverage and claudin-5 expression. Likewise, four fields from areas of white matter on each side of the brain were randomly selected and obtained at 10× magnification to determine the laminin density and five fields at 40× magnification were collected to examine vessel coverage and claudin-5 expression. 

### 4.4. Quantification of Laminin Density

The quantification of vessel density was performed according to the previously described methodology [16]. Briefly, multichannel channel images were acquired to quantify laminin density. The acquired images were examined without knowledge of the experimental groups. All images were processed using the measurement function in the software ImageJ version 1.53 (NIH, Bethesda, MD, USA). The immunoreactive-positive areas for laminin were measured, quantified, and divided by the total area of each image. Laminin density was expressed as a percent of the total area.

### 4.5. Quantification of Vascular Pericyte and Astrocyte Coverage and Claudin-5 Expression

The quantification of vascular coverage and claudin-5 expression was performed according to previously described methods [16]. Briefly, multichannel channel images were acquired to quantify vascular pericyte and astrocyte coverage and claudin-5 expression. Each wavelength was obtained separately using identical camera settings for each channel. They were then pseudo-merged into an RGB image in Adobe Photoshop and then split into either the PDGFRβ (red), GFAP (red), or claudin-5 (red) channels and the laminin (green) channel. The outer margins of the blood vessels were defined, and the total area of the blood vessels was measured by thresholding the laminin source image. The red channel (PDGFRβ, GFAP, or claudin-5) was then subtracted from the green channel (laminin) using the image calculator in ImageJ. The difference was the area of the vessels lacking coverage by pericytes or astrocytes or claudin-5 expression. The threshold used for laminin alone was also used for the new subtracted image. The vessel area without pericyte or astrocyte coverage or claudin-5 expression (green-red) was deducted from the total vessel area (green) and divided by the total vessel area (green). Vascular pericyte coverage, astrocyte coverage, or claudin-5 expression was expressed as a percent of the total surface area of the blood vessels.

### 4.6. Statistical Analysis

Values were presented as medians and scatter plots. Statistical analyses were performed using the Statistica (Dell Statistica, Tulsa, OK, USA) analysis program. The data were analyzed using a two-way analysis of variance (ANOVA), followed by Tukey’s post hoc test when a significant difference was detected. A *p* value of *p* < 0.05 was considered statistically significant.

## Figures and Tables

**Figure 1 ijms-23-04180-f001:**
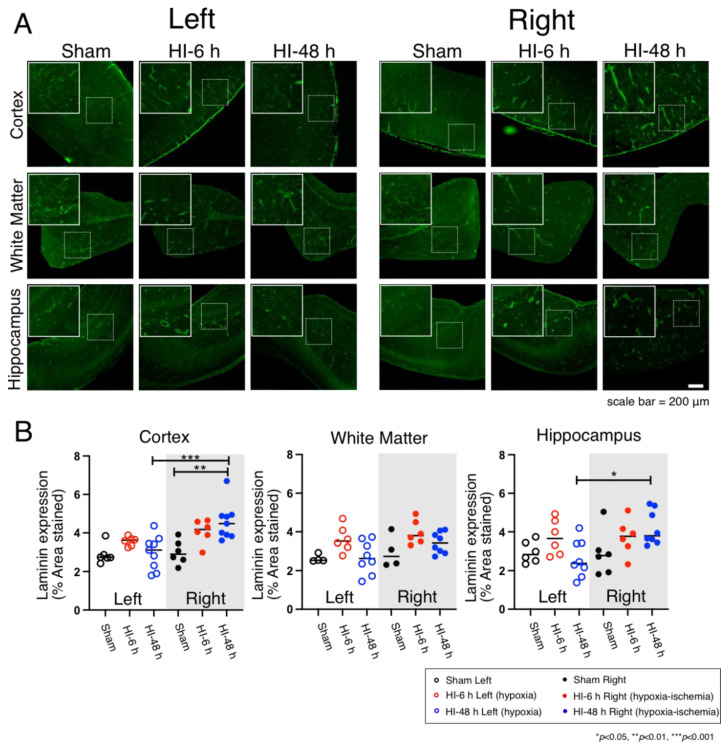
Immunohistochemical images and quantification of microvessels in the cerebral cortex, white matter, and hippocampus on the left hypoxic and right HI sides in the brain of the neonatal rats in the Sham, HI-6 h, and HI-48 h groups. (**A**) Representative images of microvessels (laminin, Green) in the cerebral cortex, white matter, and hippocampus in neonatal rats. Magnification, 10×. Each inset contains high magnification images. Scale bar = 200 μm. (**B**) Quantification of microvessel density shown by laminin expression (% Area Stained) in the cerebral cortex, white matter, and hippocampus of left hypoxic and right HI sides of rat brain in Sham (left; black open circles, right; black closed circles), HI-6 h (left; red open circles, right; red closed circles), and HI-48 h (left; red open circles, right; red closed circles). Values’ median and dot plots. * *p* < 0.05, ** *p* < 0.01, *** *p* < 0.001.

**Figure 2 ijms-23-04180-f002:**
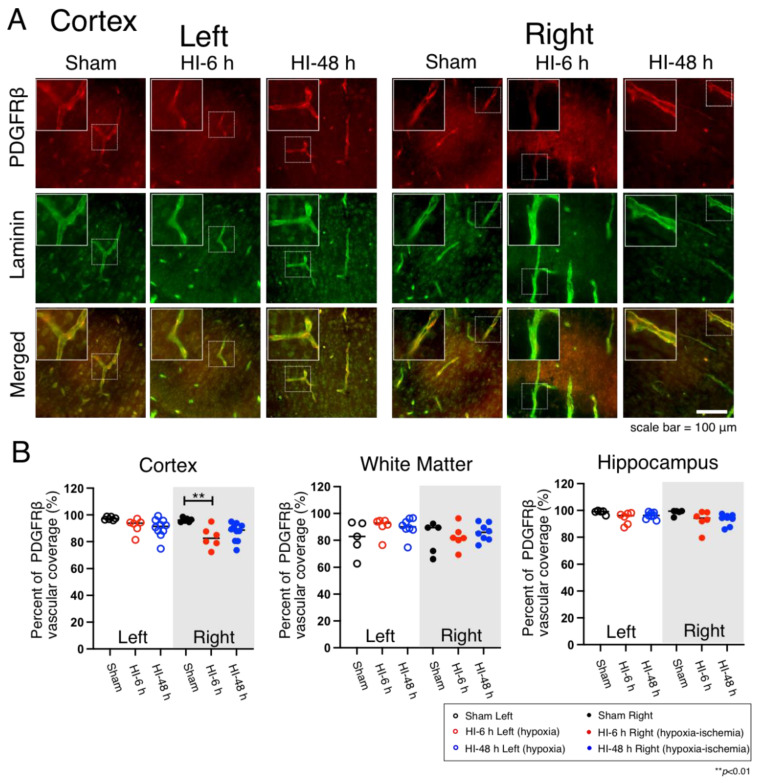
Immunohistochemical expression and quantification of pericyte coverage in the cerebral cortex, white matter, and hippocampus on the left hypoxic and right HI sides in the brain of the neonatal rats in the Sham, HI-6 h, and HI-48 h groups. (**A**) Representative images of pericyte (PDGFRβ, Red), microvessels (Laminin, Green), and merged double immunostaining in the cerebral cortex of the neonatal rats. Magnification, 40×. Each inset contains high magnification images. Scale bar = 100 μm. The immunohistochemical images in white matter and hippocampus are shown in Appendix A. (**B**) Quantification of pericyte coverage shown by percent of PDGFRβ vascular coverage in the cerebral cortex, white matter, and hippocampus of left hypoxic and right HI sides of rat brain in Sham (left; black open circles, right; black closed circles), HI-6 h (left; red open circle, right; red closed circles), and HI-48 h (left; red open circles, right; red closed circles). Values median and dot plots. ** *p* < 0.01.

**Figure 3 ijms-23-04180-f003:**
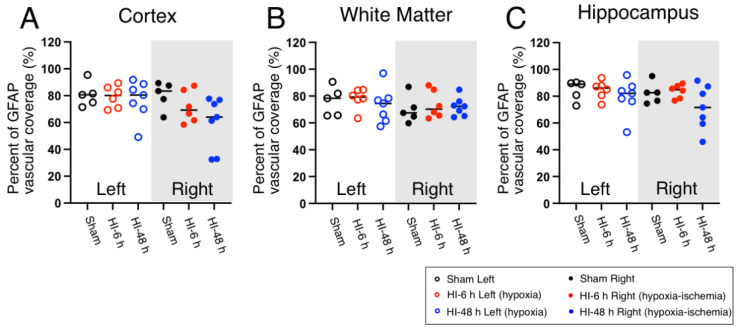
Quantification of astrocyte coverage in the cerebral cortex (**A**), white matter (**B**), and hippocampus (**C**) on the left hypoxic and right HI sides in the brain of the neonatal rats in the Sham, HI-6 h, and HI-48 h groups. Quantification of astrocyte coverage shown by percent of GFAP vascular coverage in the cerebral cortex (**A**), white matter (**B**), and hippocampus (**C**) of left hypoxic and right HI sides of rat brain in Sham (left; black open circles, right; black closed circles), HI-6 h (left; red open circles, right; red closed circles), and HI-48 h (left; red open circles, right; red closed circles). Values median and dot plots. The quantification was based upon immunohistochemical images found in Appendix A.

**Figure 4 ijms-23-04180-f004:**
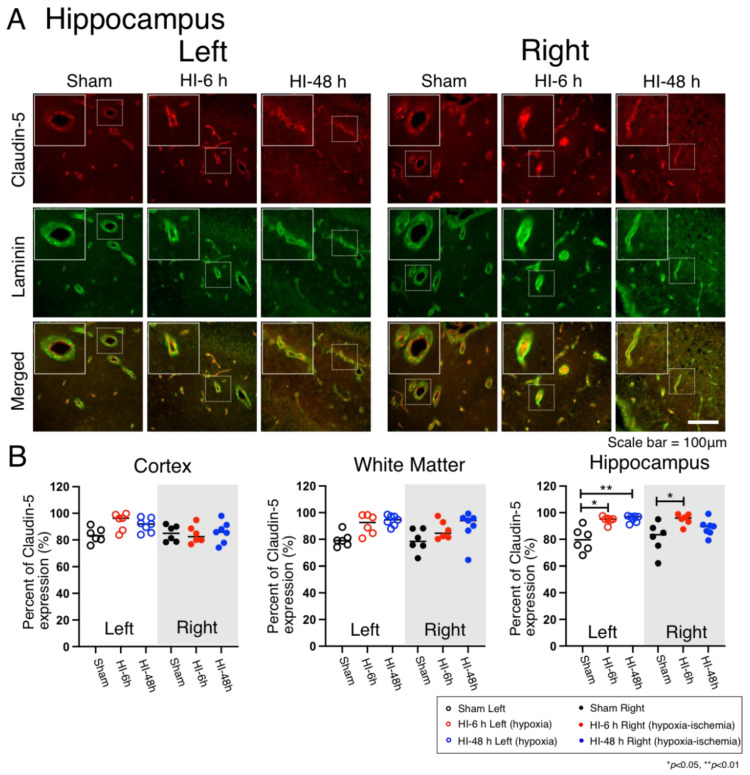
Immunohistochemical expression and quantification of vascular claudin-5 expression in the cerebral cortex, white matter, and hippocampus on the left hypoxic and right HI sides in the brain of the neonatal rats in the Sham, HI-6 h, and HI-48 h groups. *(***A**) Representative images of claudin-5 (Claudin-5, Red), microvessels (Laminin, Green), and merged double immunostaining in the cerebral cortex of left hypoxic and right HI sides of rat brain in Sham, HI-6 h, and HI-48 h groups. Magnification, 40×. Each inset contains high magnification images. Scale bar = 100 μm. The immunohistochemical images in the cerebral cortex and hippocampus are shown in Appendix A. (**B**) Quantification of vascular claudin-5 expression shown by percent of vascular claudin-5 expression in the cerebral cortex, white matter, and hippocampus of the hypoxic left and HI right sides of rat brain in Sham (left; black open circles, right; black closed circles), HI-6 h (left; red open circles, right; red closed circles), and HI-48 h (left; red open circles, right; red closed circles) groups. Values median and dot plots. * *p* < 0.05, ** *p* < 0.01.

**Figure 5 ijms-23-04180-f005:**
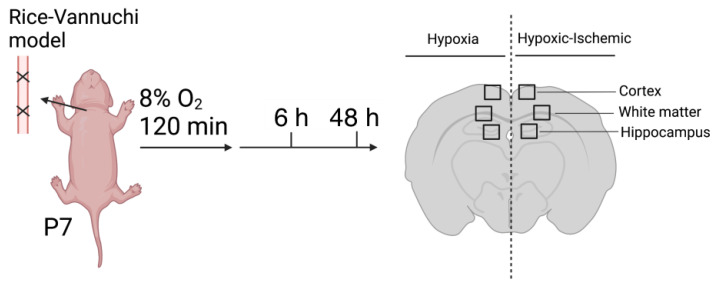
Schema of experimental design.

## Data Availability

Further information regarding the resources, reagents, and data availability should be directed to the corresponding author and will be considered upon request.

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
