# Peer review of "Time Course of Changes in the Neurovascular Unit after Hypoxic-Ischemic Injury in Neonatal Rats"

_ijms, 2022, doi:10.3390/ijms23084180_

Round 1

Reviewer 1 Report

The manuscript presented from Kazuki Hatayama et al., entitled "Time Course of Changes in the Neurovascular Unit after Hypoxic-Ischemic Injury in Neonatal Rats", is interesting. However I would suggest the authors to improve an important point:

Past and recent studies showed that in the Neurovascular Unit, neurotrophic factors as BDNF (that is expressed in neurons, astrocytes, endothelium and microglia) have a critical role in the context of Ischemic Injury. I would encourage the author to investigate by immunohistochemistry  BDNF levels in their model.

Next they should improve also the introduction and discussion of the manuscript, including BDNF.

  • Steliga, A., KowiaÅ„ski, P., Czuba, E. et al. Neurovascular Unit as a Source of Ischemic Stroke Biomarkers—Limitations of Experimental Studies and Perspectives for Clinical Application. Transl. Stroke Res. 11, 553–579 (2020). https://doi.org/10.1007/s12975-019-00744-5
  • Cacialli P. Neurotrophins Time Point Intervention after Traumatic Brain Injury: From Zebrafish to Human. International Journal of Molecular Sciences. 2021; 22(4):1585. https://doi.org/10.3390/ijms22041585

Author Response

Reviewer #1

General comments:

  • The manuscript presented from Kazuki Hatayama et al., entitled "Time Course of Changes in the Neurovascular Unit after Hypoxic-Ischemic Injury in Neonatal Rats", is interesting. However I would suggest the authors to improve an important point:

Response:  We appreciate it that the reviewer finds our manuscript is interesting.

  • Past and recent studies showed that in the Neurovascular Unit, neurotrophic factors as BDNF (that is expressed in neurons, astrocytes, endothelium and microglia) have a critical role in the context of Ischemic Injury. I would encourage the author to investigate by immunohistochemistry  BDNF levels in their model.

Response:  We have read with interest the suggested manuscripts by Steliga, A., et. al and Cacialli, P. [1, 2].  We agree that it would be very interesting to explore the changes in the Neurovascular Unit with regard to BDNF expression levels.  However, the article by Steliga, A., et. al [2] summarizes findings after stroke in adult subjects, which most likely differ considerably from those of newborns after exposure to HI related brain injury, and also refers to blood levels of BDNF as prognosticators of injury rather than their presence as components of the NVU [2]. Unfortunately, we do not have residual blood from the current study subjects.  The paper by Cacialli, P. summarized changes in levels of BDNF within the parenchymal brain tissue after traumatic brain injury in zebra fish and other species [1].   Although it would be of interest to examine the presence of BDNF and other growth factors within the NVU, our study has focused primarily on some of the structural components of the NVU.  Even though it would be of interest to examine the BDNF and other growth factor levels by immunohistochemistry, we cannot be certain that the BDNF levels could be adequately quantified within the NVU with immunohistochemical analysis.  In addition, appropriate study of this area is very complex.  Studies regarding BDNF levels would need to include mature BDNF/proBDNF and its transmembrane receptor signaling systems including TrkB and p75NTR, along with metalloproteinases, which could also be involved. Therefore, to do this analysis appropriately multiple cell sources would need to be identified including BDNF expression levels, factor-specific/receptor probes would have to be tested along with cell-specific markers using dual RNAscope methods (IHC/ISH) in order to achieve effective results for this type of analysis. Consequently, it would be a major undertaking to determine BDNF levels appropriately within the NVU.  Hence, these studies are beyond the scope of the current study.  Nonetheless, we have emphasized that BDNF and other growth factor determinations would be of interest in future studies with regard changes in the NVU with and with exposure to hypoxic-ischemic injury (Page 9, line 363).

  • Next, they should improve also the introduction and discussion of the manuscript, including BDNF.

Response:  We have added the importance of BDNF and other growth factors to the discussion (Page 9, line 363).

References

  1. Cacialli, P., Neurotrophins Time Point Intervention after Traumatic Brain Injury: From Zebrafish to Human. Int J Mol Sci 2021, 22, (4).
  2. Steliga, A.; Kowianski, P.; Czuba, E.; Waskow, M.; Morys, J.; Lietzau, G., Neurovascular Unit as a Source of Ischemic Stroke Biomarkers-Limitations of Experimental Studies and Perspectives for Clinical Application. Transl Stroke Res 2020, 11, (4), 553-579.

Reviewer 2 Report

Hatayama and colleagues performed a hypoxic-ischemic injury in P7 rats to study the changes of the neurovascular unit by using immunohistochemistry. Their goal was to assess the vascular density, pericyte coverage of the vessels, astrocytic coverage of the vessels, and expression of the TJ protein claudin-5 in 3 brain regions. The methods are clearly explained.

1-I am concerned with the laminin staining that clearly labels a lot more than just vessels. It looks like neurons are also stained with the antibody used in this study, it is visible in all the figures. To my understanding, the way the vessel density was quantified reflects the density of laminin staining, and since neurons are also stained, the results do not represent the vessel density only. Moreover, many of the representative pictures are not convincing to me. For example, in Fig1 the quantification shows more laminin density between left HI48h and right HI48h but the pictures do not reflect the quantification. The pictures shown in supplemental figures are not necessarily concordant with the results from figure 1.

2-GFAP staining is not ideal for all brain regions. I would suggest using S100b for cortical regions (see article by Zhang et al., 2019: "Glial fibrillary acidic protein (GFAP) is the most commonly used astrocytic marker, but as the major intermediate filament composing cytoskeleton, GFAP immunolabeled only about 15% of the total astrocyte volume [6], and more than 40% of astrocytes were found to be GFAP-negative in the adult rat hippocampus [7]." https://doi.org/10.1155/2019/9605265). Therefore, the results obtained with GFAP staining in the cortex don't allow for the quantification of the astrocyte vessel coverage.

3-In order to improve rigor and reproducibility, it would be good to include the references/catalog numbers of the antibodies that were used in this study.

Author Response

Reviewer #2

General comments:

  • Hatayama and colleagues performed a hypoxic-ischemic injury in P7 rats to study the changes of the neurovascular unit by using immunohistochemistry. Their goal was to assess the vascular density, pericyte coverage of the vessels, astrocytic coverage of the vessels, and expression of the TJ protein claudin-5 in 3 brain regions. The methods are clearly explained.

Response:  We are pleased that the reviewer has found our methods clearly explained.

  • I am concerned with the laminin staining that clearly labels a lot more than just vessels. It looks like neurons are also stained with the antibody used in this study, it is visible in all the figures. To my understanding, the way the vessel density was quantified reflects the density of laminin staining, and since neurons are also stained, the results do not represent the vessel density only. Moreover, many of the representative pictures are not convincing to me. For example, in Fig1 the quantification shows more laminin density between left HI48h and right HI 48h but the pictures do not reflect the quantification. The pictures shown in supplemental figures are not necessarily concordant with the results from figure 1.

Response:  We appreciate the reviewer’s comments.  Consequently, we have consulted Dr. Daniela Virgintino, who is an expert in immunohistochemistry with whom we have collaborated in the past [3].  She and others have previously utilized the same or similar anti-laminin antibodies to identify blood vessels in the brain of rodents and other species including humans [4-9].  We have added this point in the discussion (Page 7, line 251).  Dr. Daniela Virgintino reviewed our images and agrees that most of the staining appears to be linear in nature and therefore, mostly representative of blood vessels.

The neuron/nerve fibers staining actually reflects an aldehyde fixative induced auto-fluorescence that we were not able to avoid because we required a long fixation time interval to adequately preserve the NVU’s cellular and subcellular structures. On one hand this background was helpful to see the different areas in the sections, as green on the single channel image and a mud yellow on the overlay images. However, it is true that this could affect the fluorescence quantitation.  Unfortunately, formaldehyde aldehyde-induced fluorescence is worse on neural tissue than on other tissues and is not reduced by the highly specific monoclonal primary antibody.  Protocols that reduce fixative-induced fluorescence (e.g., aldehyde blocking methods with sodium borohydride) also reduce significant fluorescence and nerve tissue structural preservation in our experience.  We have added this point in the limitations of our study (Page 10, line 377).

Moreover, we have re-reviewed our immunohistochemical images and attempted to select images that reflect more closely the quantification of the images.  The supplemental figures were taken by 40x magnification and focused on the pericyte, astrocyte coverage and claudin-5 expression. We have replaced supplementary figure 2 to simulate more closely the mean quantification values in figure 3. They do not necessarily reflect the result of vessel density quantification because they are the higher magnification images than in figure 1.

Moreover, it should be noted that it is somewhat difficult to precisely match the quantified values because the values in the graphs represent mean values and no one animal might exactly represent these mean values.  In addition, it should be emphasized that all of the image analyses were performed without knowledge of the study groups and that our findings in the neonatal rats exposed to HI are consistent with our previous findings in fetal sheep after exposure to ischemia with reperfusion, in which we had used collagen IV as a vascular marker [3].

  • GFAP staining is not ideal for all brain regions. I would suggest using S100b for cortical regions (see article by Zhang et al., 2019: "Glial fibrillary acidic protein (GFAP) is the most commonly used astrocytic marker, but as the major intermediate filament composing cytoskeleton, GFAP immunolabeled only about 15% of the total astrocyte volume [6], and more than 40% of astrocytes were found to be GFAP-negative in the adult rat hippocampus [7]." https://doi.org/10.1155/2019/9605265). Therefore, the results obtained with GFAP staining in the cortex don't allow for the quantification of the astrocyte vessel coverage.

Response:  Many published studies have previously used GFAP as a marker for astrocytes including our own study [7, 10]. Moreover, we have previously used the identical methods to quantify astrocyte coverage in fetal sheep cerebral vasculature with and without exposure to LPS and were able to detect differences in coverage between the groups with decreased astrocytic vascular coverage in white matter and a similar non-significant trend in the cerebral cortex of the fetal sheep [10]. Although NDRG2 and S100b might be considered more suitable as labels for the distribution and number of astrocytes in the cortex and thalamus in some studies [11], these studies were performed in one month, adult and aged mice.  Therefore, if we considered using these markers for astrocytes, we would first need to compare the validity of these markers to GFAP in normal neonatal rats and in those exposed to HI in the different brain regions. This would be an extensive undertaking, which is beyond the feasibility for the current study. Furthermore, in our study we were quantifying vascular coverage by astrocytes not the total number or overall distribution of astrocytes as previously reported [11]. In addition, others consider that S100B is only expressed by a subtype of mature astrocytes that ensheath blood vessels and also by NG2-expressing cells [12, 13]. Therefore, there may not be a single optimal marker for astrocytes under all conditions, developmental ages, and brain regions [14]. Consequently, we have expanded our discussion regarding the potential for additional detailed analysis of different markers for astrocytes based upon relevant literature [11-14] but are not able to perform these extensive analyses for the current study. However, we propose that this is an excellent avenue for future study (Page 9, line 310).

  • In order to improve rigor and reproducibility, it would be good to include the references/catalog numbers of the antibodies that were used in this study.

Response:  We have added the catalog numbers for the antibodies that we used in the study to the methods (Page 12, line 463).

References

3. Virgintino, D.; Girolamo, F.; Rizzi, M.; Ahmedli, N.; Sadowska, G. B.; Stopa, E. G.; Zhang, J.; Stonestreet, B. S., Ischemia/Reperfusion-induced neovascularization in the cerebral cortex of the ovine fetus. J Neuropathol Exp Neurol 2014, 73, (6), 495-506.

  1. Eriksdotter-Nilsson, M.; Björklund, H.; Olson, L., Laminin immunohistochemistry: a simple method to visualize and quantitate vascular structures in the mammalian brain. J Neurosci Methods 1986, 17, (4), 275-86.
  2. Krueger, M.; Hartig, W.; Reichenbach, A.; Bechmann, I.; Michalski, D., Blood-brain barrier breakdown after embolic stroke in rats occurs without ultrastructural evidence for disrupting tight junctions. PLoS One 2013, 8, (2), e56419.
  3. Bertossi, M.; Virgintino, D.; Errede, M.; Roncali, L., Immunohistochemical and ultrastructural characterization of cortical plate microvasculature in the human fetus telencephalon. Microvasc Res 1999, 58, (1), 49-61.
  4. El-Khoury, N.; Braun, A.; Hu, F.; Pandey, M.; Nedergaard, M.; Lagamma, E. F.; Ballabh, P., Astrocyte end-feet in germinal matrix, cerebral cortex, and white matter in developing infants. Pediatr Res 2006, 59, (5), 673-9.
  5. Natah, S. S.; Srinivasan, S.; Pittman, Q.; Zhao, Z.; Dunn, J. F., Effects of acute hypoxia and hyperthermia on the permeability of the blood-brain barrier in adult rats. J Appl Physiol (1985) 2009, 107, (4), 1348-56.
  6. Girolamo, F.; Errede, M.; Longo, G.; Annese, T.; Alias, C.; Ferrara, G.; Morando, S.; Trojano, M.; Kerlero de Rosbo, N.; Uccelli, A.; Virgintino, D., Defining the role of NG2-expressing cells in experimental models of multiple sclerosis. A biofunctional analysis of the neurovascular unit in wild type and NG2 null mice. PLoS One 2019, 14, (3), e0213508.
  7. Disdier, C.; Awa, F.; Chen, X.; Dhillon, S. K.; Galinsky, R.; Davidson, J. O.; Lear, C. A.; Bennet, L.; Gunn, A. J.; Stonestreet, B. S., Lipopolysaccharide-induced changes in the neurovascular unit in the preterm fetal sheep brain. J Neuroinflammation 2020, 17, (1), 167.
  8. Zhang, Z.; Ma, Z.; Zou, W.; Guo, H.; Liu, M.; Ma, Y.; Zhang, L., The Appropriate Marker for Astrocytes: Comparing the Distribution and Expression of Three Astrocytic Markers in Different Mouse Cerebral Regions. Biomed Res Int 2019, 2019, 9605265.
  9. Deloulme, J. C.; Raponi, E.; Gentil, B. J.; Bertacchi, N.; Marks, A.; Labourdette, G.; Baudier, J., Nuclear expression of S100B in oligodendrocyte progenitor cells correlates with differentiation toward the oligodendroglial lineage and modulates oligodendrocytes maturation. Mol Cell Neurosci 2004, 27, (4), 453-65.
  10. Hachem, S.; Aguirre, A.; Vives, V.; Marks, A.; Gallo, V.; Legraverend, C., Spatial and temporal expression of S100B in cells of oligodendrocyte lineage. Glia 2005, 51, (2), 81-97.

14.      Wang, D. D.; Bordey, A., The astrocyte odyssey. Prog Neurobiol 2008, 86, (4), 342-67

Round 2

Reviewer 1 Report

The authors satisfied all my concerns. I propose the acceptance in the present form

Reviewer 2 Report

I appreciate the fact that the authors discussed the limitations due to the laminin staining and the astrocyte staining.